# Effect of Increased Ionic Liquid Uptake via Thermal Annealing on Mechanical Properties of Polyimide-Poly(ethylene glycol) Segmented Block Copolymer Membranes

**DOI:** 10.3390/molecules26082143

**Published:** 2021-04-08

**Authors:** Gokcen A. Ciftcioglu, Curtis W. Frank

**Affiliations:** 1Department of Chemical Engineering, Marmara University, Istanbul 34722, Turkey; 2Department of Chemical Engineering, Stanford University, Stanford, CA 94305, USA; cwfrank@stanford.edu

**Keywords:** copolymerization, bicontinuous structures, ionic liquids, polymer electrolyte membranes

## Abstract

Proton exchange membranes (PEMs) suffer performance degradation under certain conditions—temperatures greater than 80 °C, relative humidity less than 50%, and water retention less than 22%. Novel materials are needed that have improved water retention, stability at higher temperatures, flexibility, conductivity, and the ability to function at low humidity. This work focuses on polyimide-poly(ethylene glycol) (PI-PEG) segmented block copolymer (SBC) membranes with high conductivity and mechanical strength. Membranes were prepared with one of two ionic liquids (ILs), either ethylammonium nitrate (EAN) or propylammonium nitrate (PAN), incorporated within the membrane structure to enhance the proton exchange capability. Ionic liquid uptake capacities were compared for two different temperatures, 25 and 60 °C. Then, conductivities were measured for a series of combinations of undoped or doped unannealed and undoped or doped annealed membranes. Stress and strain tests were performed for unannealed and thermally annealed undoped membranes. Later, these experiments were repeated for doped unannealed and thermally annealed. Mechanical and conductivity data were interpreted in the context of prior small angle X-ray scattering (SAXS) studies on similar materials. We have shown that varying the compositions of polyimide-poly(ethylene glycol) (PI-PEG) SBCs allowed the morphology in the system to be tuned. Since polyimides (PI) are made from the condensation of dianhydrides and diamines, this was accomplished using components having different functional groups. Dianhydrides having either fluorinated or oxygenated functional groups and diamines having either fluorinated or oxygenated diamines were used as well as mixtures of these species. Changing the morphology by creating macrophase separation elevated the IL uptake capacities, and in turn, increased their conductivities by a factor of three or more compared to Nafion 115. The stiffness of the membranes synthesized in this work was comparable to Nafion 115 and, thus, sufficient for practical applications.

## 1. Introduction

Nafion is a widely used state-of-the-art proton exchange membrane (PEM). Moreover, there are other PEMs for fuel cell applications based on perfluorosulfonic acid (PFSA) derived from Nafion membrane such as Nafion 115, Nafion 117, etc. [1,2,3,4,5]. Nafion has been popular due to its unique properties such as flexibility, thermal and chemical stability, and conductivity when fully hydrated. However, conductivity dramatically depends on the water content in the membrane and drops by orders of magnitude at lower levels of relative volatility (RV). This inherent limitation of Nafion results from the lack of long-range and ordered nano- or meso-porous characteristic morphology, leading to conductivity change [5]. Considerable effort has been directed toward water management as well as toward finding novel materials that have the potential to be used as proton exchange membranes with superior properties [6,7,8,9].

The objective of most work in the literature has been to find the optimal membrane that can operate at higher temperatures and at low humidity, which will minimize the expense of the essential water management technology for Proton Exchange Membrane Fuel Cells (PEMFCs) [10]. There has been significant effort in recent years to develop PEMs that can operate efficiently at temperatures higher than 80 °C and low relative humidity [11,12,13,14,15,16,17,18,19,20,21,22,23]. Many attempts have been made to modify the structure of conventional Nafion membranes to improve their operational properties, e.g., by incorporating hygroscopic inorganic nanoparticles like SiO_2_, TiO_2_, and ZrO_2_ [24,25,26]. There have also been extensive studies to synthesize entirely different materials from block or random copolymers, e.g., polystyrene sulfonate-block-polymethyl butylene (PSS-b-PMB) copolymers [11], chitosan/phosphotungstic acid [27], polyimides (PIs), and polyethylene glycol (PEG)-containing PI [28,29,30,31,32,33,34,35,36,37,38,39,40,41,42,43,44,45,46,47].

Ionic liquid-incorporated membranes have also been studied in recent research due to their high ion conductivity, high thermal and electro-chemical stability, and ionic behavior at a variety of pH levels [48,49,50,51,52,53,54,55]. It has been shown that the incorporation of ionic liquids affected the structure and enhanced the conductivity [28,54,55]. For example, Sekhon et al. used ionic liquids as anhydrous ion vehicles and doped sulfonated poly (ether ketone), leading to an enhancement of the conductivity [54]. Deligöz et al. developed a highly conductive and thermomechanically stable complex membrane based on sulfonated polyimide/ionic liquids (IL), for which the dynamic mechanical analysis indicated only a slight decrease in the mechanical strength until the temperature reached 350 °C [55]. These studies clearly show that material characteristics such as structural order, intermolecular interactions, and free volume play critical roles in the design of membranes with desirable properties. Membranes derived from aromatic diamines or dianhydrides, the subject of the present work, are shown to have such characteristics [28].

Recent studies showed that segmented block copolymer (SBC) membranes derived from PI and PEG doped with ionic liquid ethylammonium nitrate (EAN) increased the conductivity by two to five times upon thermal annealing at around 100–140 °C [31]. It was shown that the ionic liquid also acted as a plasticizing agent, which enhanced polymer chain mobility, thus improving the membrane ionic transport capability by decreasing the intermolecular forces among polymer chains [31]. 

The goal of this study is to show how the conductivity of membranes can be modified via thermal annealing as a function of increasing IL uptake (ILU). It is known that the incorporation of ionic liquids will enhance the conductivity and electrochemical stability of membranes by affecting their free volume and flexibility [51,52]. The primary focus of this work has been to understand the conductivity properties of 15 different PI-PEG SBC membranes doped with two different ionic liquids. The information gained should serve to elucidate the effect of the changes in the transport properties of membranes as a function of ILU and the changes in the mechanical properties induced by thermal annealing.

## 2. System Design

Our research was motivated by the need for a material that can work under harsher environments than Nafion and its derivatives. As mentioned in many previous studies [28,29,30,31], challenges remain to overcome issues like needing to work at low operating temperature, requiring high humidity, or high-water absorption that may cause swelling and mechanical instability. Thus, the objective is to find optimal materials that can effectively respond to all or a portion of these challenges.

The PEM community has learned that molecular level identity and interactions greatly impact the bulk behavior of the corresponding polymer [28,29,30,31]. Many studies have been carried out to understand the effect of nanometer-level structures or polymer–liquid interactions on PEM material properties. Some specific polymers that have been examined include: polyethylene oxide/poly(methyl methacrylate) [56], polytetrafluoroethylene and polyethylene [57], poly(ethyleneimine) [58], sulfonated poly(etherketone) [59,60], poly(styrene-b-methyl methacrylate) [61], polyisoprene–polystyrene–poly(ethylene oxide) [53], polystyrenesulfonate–polymethylbutylene and polystyrene–poly(ethylene oxide) [62], poly(vinylidene fluoride) [63], poly(vinyl chloride), cellulose triacetate [64], poly(n-isopropylacrylamide) [65,66,67,68], polysulfides [69], polybutadiene [70], poly(styrene-b-vinylpyridine) [71,72], poly(vinylpyridine) [73,74], poly(vinyl acetate) [75], poly(propylene carbonate) [76,77,78,79], polysulfone [80], polyurethane [81], and polyimides [55,82,83,84,85].

In recent studies of ionic liquid uptake by PEG containing-PI SBC membranes done by Coletta et al., it was demonstrated that there is a direct link between structure, conductivity, composition, and polymer–ionic liquid interactions [28,29,30]. Coletta et al. used prior small angle X-ray scattering (SAXS) to understand the effect of changing aromatic dianhydrides or diamines within the copolymer membrane. First, undoped and ethylammonium nitrate (EAN)-doped PEG1500 (50 wt%)-4,4’-(hexafluoroisopropylidene) diphthalic anhydride (6FDA)- 2,2’-bis (4-aminophenyl) hexafluoropropane (AP6F) and PEG1500 (50 wt%)-4,4’-oxydiphthalic anhydride (ODPA)-AP6F membranes were compared to understand the effect of changing the aromatic dianhydride. SAXS experiments were evaluated comprehensively. For the undoped case, there were relatively few structural features. The undoped ODPA material showed a shoulder that is consistent with randomly distributed PEG domains in the polymer with a PEG correlation length of 0.9 nm or a radius of 1.6 nm. In addition, a broad peak was observed that was attributed to aromatic polyimide ordering, with a spacing value of 15.3 nm. On the other hand, the undoped 6FDA material showed only a single feature, also a shoulder consistent with randomly distributed PEG domains, with a correlation length of 0.9 nm, corresponding to a radius of 1.6 nm. It was also noted that there were noticeable changes in polymer structure when comparing the undoped and doped polymer systems in both systems. The doped ODPA material showed a more ambiguous shoulder feature with an estimated spacing value at 15.0 nm for the aromatic polyimide order and a spacing value of 11.6 nm for the PEG order. The doped 6FDA material also presented a higher spacing value of 13.4 nm for PEG order, and it was found to be relatively broader in shape compared to the ODPA-containing material. In summary, both doped materials showed the presence of a peak that was attributed to the spacing between ionic liquid-doped PEG domains [28]. 

In addition to studying the morphology of undoped and EAN-doped PEG1500 (50 wt%)-6FDA-AP6F and PEG1500 (50 wt%)-ODPA-AP6F, Coletta et al. determined ionic liquid uptakes and conductivities and showed that an ether-containing (ODPA) dianhydride membrane had a lower uptake value and, thus, lower conductivity compared to a fluorinated material. This lower uptake value was correlated with the fact that SAXS data for the 6FDA system showed less rigidity (less defined ordering). In the same study, the effect of changing aromatic diamines (PEG1500 (50 wt%)-6FDA-AP6F and PEG1500 (50 wt%)-6FDA- 4,4’-(1,3-phenylenedioxy) dianiline (PDODA)) was also studied. For the undoped case, both materials presented almost similar structures, showing randomly distributed PEG domains. The PDODA-containing membrane presented aromatic regularity with a spacing value of 2.2 nm, which was attributed to the miscibility changes or steric interference from polyimide order. 

Analysis of EAN-doped membranes also involved a comparison of SAXS signatures. Both systems showed a single peak attributed to PEG-EAN domains. The primary structural difference was the relative broadness and definition of the peak, with the PDODA system showing more definition. Coletta et al. used full-width-half-maximum (FWHM) calculations to quantify the differences in the SAXS features. Especially when the SAXS data showed a similar output, quantifying the features helped to understand the behavioral change seen in ionic liquid uptakes and conductivity values. It was shown that the fluorinated-based membrane had a more defined PEG-EAN order, where FWHM was found to be around 0.42 and 0.51 nm^−1^ for PDODA- and AP6F-containing materials, respectively. A larger FWHM indicates a higher possibility of ionic liquid uptake, where the ether-containing material had a dominant aromatic phase order with less defined PEG-EAN domains holding a possibility of lower uptake value. To further illustrate the relevance of this result, an ILU study showed that the fluorinated-based material had a higher value of ILU by a factor of two. 

An understanding of these structure-property analyses is important to be able to synthesize copolymers with desirable properties. Coletta et al. showed that the conductivities of the membranes are affected by the amount of ionic liquid uptake, as well as free volume and rigidity. In general, more free volume means more ionic liquid uptake and less rigidity; thus, one would expect higher ion transport, but this is likely to lead to mechanically weaker membranes [28,29,30].

The same research group also examined how the chemical and mechanical properties can be affected when containing different amounts of PEG-EAN domains. Woo et al. studied varying compositions of PEG1500 between 0 to 46.8 wt% in the PEG1500-6FDA-PDODA SBC membrane system [31]. In this study, a series of SAXS experiments on materials with changing PEG contents was conducted for undoped and doped cases. These analyses were repeated for different temperatures, changing from 25 to 140 °C for material containing 46.8 wt% PEG1500. 

Woo et al. first showed that for the undoped case, changing PEG concentration in the materials only yielded very weak broad shoulders with q values of 0.05–0.2 A^−1^, which were attributed to weak PEG-domains [31]. However, with EAN doping, the structural change became observable and PEG contents higher than 26.2 wt% showed more prominent peaks. In summary, it was shown that increasing PEG1500 concentration caused more well-defined PEG-EAN domains to be obtained. These differences in the SAXS peaks also indicate that structural changes are resulting from domains being swollen by the EAN. 

In addition, Woo et al. explored thermal processing in an attempt to enhance the material properties. For this purpose, 46.8 wt% PEG1500 was thermally annealed for 10 min over the temperature range of 100–140 °C. From the SAXS plots for the increasing temperature of the EAN-doped SBC membrane, increasing distinctive peaks were obtained. It was noted that the SAXS behavior resembles the small-angle neutron scattering (SANS) profiles obtained for a disordered bi-continuous microemulsion. Significantly, Woo et al. proposed that the PEG-containing PI copolymers are disordered bi-continuous phase-separated structures [31]. Also, it was shown that the conductivity of IL-doped SBC membranes was increased by two to five times after thermal annealing. These results were consistent with the structural changes observed in the SAXS measurements. 

In light of these findings, the effects of adding different functionalities of dianhydride or diamine were further studied. The main reason for incorporating different types of dianhydrides or diamines was to increase IL uptake, reduce water absorption, and increase conductivity. Our approach to accomplishing this was to prepare membranes with reduced swelling, but ones in which we took advantage of control over the disordered bi-continuous phase-separated PEG domains, as suggested by Woo et al. [31]. It should be emphasized that the nature of the ionic pathways in block copolymer systems has a major impact on conductivity [28,53,86]. Thus, we have synthesized SBCs with varying compositions in order to influence the segmented bi-continuous phase-separated domains, which are directly related to the ionic liquid uptake, and thus to the conductivity and mechanical integrity.

The studies of Coletta et al. and Woo et al. demonstrate that the properties of SBCs are critically dependent on the chemical moieties incorporated into their structures. So, they have studied various PEG-containing PI membranes and investigated how these materials can gain better mechanical properties by influencing the structure. The membranes they have synthesized are summarized in Table 1. Taking into account their effort, the study was taken a step further to investigate more complex systems having multiple diamines or dianhydrides, or even a mixture of all these materials. Thus, the comparison of the PI-PEG SBC membranes synthesized in this work is presented with the synthesized membranes studied previously by Coletta et al. and Woo et al. in Table 1 below. 

The membranes synthesized in this study to elaborate on the materials of Coletta et al. and Woo et al. [31] can be divided into three major groups:Group 1: Membranes Proton Exchange Membrane_1 (PEM_1) through Proton Exchange Membrane_5 (PEM_5)Group 2: Membranes Proton Exchange Membrane_6 (PEM_6) through Proton Exchange Membrane_10 (PEM_10)Group 3: Membranes Proton Exchange Membrane_11 (PEM_11) through Proton Exchange Membrane_15 (PEM_15)

Table 2 shows the weight percentages of all materials incorporated into the SBC membranes. The chemical structures of all these materials can also be seen in Figure 1. 

Note that the only difference between the Group 1 and Group 2 membranes is the replacement of oxygenated PDODA with the same amount of fluorinated AP6F. Group 3 contains PEM_11 through PEM_15. These membranes differ from those of the first two groups (Group 1 and Group 2) in that both kinds of diamines (PDODA and AP6F) of equal amounts are incorporated into the structure of membranes, as shown in Table 2. In all these three groups (a total of 15 different SBCs), notice that different amounts of fluorinated and oxygenated dianhydrides (6FDA and ODPA) were added.

These three groups are related as being a part of PI-PEG family membranes; however, they differ from the compositional point of view. In unpublished SAXS measurements and mechanical test results, we observed that the segmented bi-continuous phase-separation and tensile strength are affected by the amount of PEG content; thus, the optimum value was selected as 42.10 wt% for the present study [28,29,30,31].

There are several motivations for designing these three major groups. The first is to observe the effect of changing the dianhydride. Thus, in all three groups, dianhydrides were changed from having only fluorinated dianhydride to only oxygenated dianhydride by systematically decreasing the content of 6FDA, while increasing the content of ODPA. Dianhydrides were selected in this manner because it was believed that the structural strength was primarily controlled by this material. Second, to see the effect of changing the diamine, the content of the diamine in Group 2 was changed to AP6F from PDODA and equally mixed in Group 3. The third reason is based on the decision of incorporating two different dianhydrides/diamines into the SBC matrix based on hydrophilic/hydrophobic balances. It was believed that changing these balances would be expected to affect the PEG phase separation and probably the water uptake values. The question of the influence of the ionic liquid uptake was more difficult to address, a priori, which made this a central element to pursue.

## 3. Experimental

### 3.1. Materials

N,N-dimethylacetamide (DMAc), 4,4’-(1,3-phenylenedioxy) dianiline (PDODA) (oxygenated based diamine), 2,2’-bis (4-aminophenyl) hexafluoropropane (AP6F) (fluorinated diamine), 4,4’-oxydiphthalic anhydride (ODPA) (oxygenated dianhydride or ether-containing dianhydride), 4,4’-(hexafluoroisopropylidene) diphthalic anhydride (6FDA) (fluorinated dianhydride), poly (ethylene glycol) bis (3-aminopropyl) terminated (PEG1500)—number indicating the molecular weights of the PEG diamines, were purchased from Sigma-Aldrich. The ionic liquids ethylammonium nitrate (EAN (>97%)) and propylammonium nitrate (PAN (>97%)) were purchased from Iolitec. All the materials were used as received.

### 3.2. Synthesis of PEG-Containing Poly(amic acid) Solution

A two-step production method was conducted to produce free-standing polyimide membranes. First, PEG1500 and one of the aromatic diamines (PDODA or AP6F) and a series of combinations of the two aromatic diamines (PDODA and AP6F) were mixed. Then, DMAc solvent was added, and the system was heated gently under nitrogen until all the contents were dissolved. 

After cooling the mix to room temperature, one of the solid aromatic dianhydrides (6FDA or ODPA) or a series of combinations of the mixture of two aromatic dianhydrides (6FDA and ODPA) were incorporated with stirring over a period of 60 min. The resulting mixture then continued to be stirred under nitrogen for 24 h.

### 3.3. Imidization of the Poly(amic acid) Solution to Produce PEG-PI Membrane

After mixing the solution for 24 h, it was poured into a Teflon dish and thermally imidized in an oven following the standard protocol: heat the sample from room temperature to 90 °C for 2 h and 15 min, then ramp to 130 °C over 3 h, hold at 130 °C for 11 h, heat to 155 °C over 3 h, hold at 155 °C for 1 h, and finally cool the sample to 25 °C for the 4 h time period. 

The chemical structures of the materials used in the two-step production method of PI-PEG SBC membranes are presented in Figure 1.

### 3.4. Characterization

Fourier transform infrared spectroscopy (FTIR), thermal gravimetric analysis (TGA), differential scanning calorimetry (DSC), and conductivity measurements were performed for the characterization of the PEMs listed in Table 2.

FTIR measurements were conducted using a Nicolet iS50 FT/IR Spectrometer at room temperature. Measurements were first performed on the poly (amic acid) precursors and then on the resulting membranes of Table 2 after thermal imidization. 

TGA was used to determine the thermal stabilities of the systems by measuring their weight changes using the Texas Instrument Q500 system. Membrane samples between 8 and 12 mg were loaded into alumina pans for testing. The samples were heated according to a ramping protocol from 25 to 600 °C at a rate of 10 °C/min, and mass loss was recorded. 

DSC analyses were conducted using a TA Instrument Q2000 for all the free-standing membranes produced. For DSC analysis, samples between 8 and 12 mg were loaded into aluminum pans and then treated according to the following thermal protocol: equilibrate at 20 °C, ramp at 5 °C/min up to 120 °C, hold at 120 °C for 2 min, ramp at 5 °C/min to 20 °C, hold at 20 °C for 2 min, and then repeat the cycle two more times.

### 3.5. IL Incorporation and Water Uptake

Appropriately sized (3 ± 2 cm × 0.5 ± 0.2 cm) sample films were cut from PI-PEG SBC membranes of Table 2 and then placed into EAN and PAN for IL incorporation or into the water for water uptake (WU) using a closed container at room temperature for 24 h. IL uptake (ILU) and WU experiments were conducted on a subset of measurements by weighing undoped and doped membranes after being soaked for 1 h in EAN, PAN, or water. Three samples of equal shape and size were used for each of the 45 different measurements for determining ILU and WU properties of the PEMs shown in Table 2 at a given temperature. The respective ILU and WU properties are reported as the average of these three measurements. These experiments were conducted at two temperature levels: 25 and 60 °C.

### 3.6. Electrochemical Impedance Spectroscopy (EIS) Measurements and Membrane Proton Conductivity 

Electrochemical impedance (EI) is the response of an electrochemical system (cell) to an externally applied potential. We used the Autolab potentiostat/galvanostat PGSTAT204 instrument combined with the FRA32M electrochemical impedance spectroscopy (EIS) module to measure the potentiostatic impedance of the cell containing the membranes of Table 1 as electrolyte over a wide frequency range of 0.1 Hz to 2 MHz, with 10 points per decade. The amplitude of externally applied alternating current (AC) voltage was set to be 0.01 V and the current range of 100 nA to 1A was selected for the frequency scan. The measured real and imaginary parts of the complex impedances of the systems were then represented by Nyquist plots. These Nyquist plots were also modeled by using the calculated EIS response of an equivalent circuit for further validation. In search of proper equivalent circuits for systems where doped membranes were used as an electrolyte, the best fit was found by replacing the Faradaic impedance of a double layer, i.e., the interface between the electrode and the neutral bulk electrolyte, by the appropriate constant phase element that takes into account the deviation of the double-layer capacity from ideal behavior. The intersection of the Nyquist curve with the real axis at the high frequency is used to calculate the proton conductivity of the membrane electrolyte for doped membranes. Then, after evaluating the resistivity for all cases, where the resistivity is obtained from the measured Nyquist plot and compared with the corresponding fit value of the equivalent electrical circuit, Equation (1) was used to calculate the conductivities [56]:σ = L/(RA)(1)
where σ is the proton conductivity in mS.cm^−1^, L is the length of the membrane in cm, R is the resistivity of the membrane in Ω, and A is the membrane area in cm^2^ found using width and thickness of the membrane. Three different samples were prepared for each set of experiments. EIS measurements and equivalent circuit simulations were performed for each case.

### 3.7. Mechanical Testing

The responses of the membranes of Table 2 to the externally applied force were measured with a strain rate of 5 cm/min on an Instron 5500 machine, with the stress/strain curves plotted. The elastic modulus and yield stress at the maximum force of the membrane were also measured. The samples used in the measurements were prepared with dimensions of 20 ± 5 mm × 5 ± 2 mm. Mechanical strength tests were conducted for both untreated and thermally treated membranes. 

## 4. Results and Discussion

In Figure 2, the picture of the final product PEM_3 is shown and compared with Nafion 115 using the non-mechanical folding test. It can be seen from this figure that the membrane is very robust and easy to handle; in other words, it appears to have sufficient stability for practical usage, and it is also as flexible as Nafion 115. Flexibility is an important property to achieve for easy handling without ruptures. Thus, in the sample shown in Figure 2, the PEM_3 membrane is as flexible as Nafion 115. This is believed to be achieved as a result of the incorporation of PEG1500 into the structure. 

In our work, we hypothesized that the mechanical strength of the polymers could be enhanced through systematic variation of the identity and concentration of several dianhydrides and diamines. Thus, fifteen different membranes were synthesized: In Group 1 (PEM_1 to PEM_5), the diamine component is only oxygenated-based, in Group 2 (PEM_6 to PEM_10), the diamine component is only fluorinated-based, and in Group 3 (PEM_11 to PEM_15), the diamine component is a mixture of equal amounts of oxygenated- and fluorinated-based.

### 4.1. FTIR Measurements

To confirm the thermal imidization of the polyimide membranes listed in Table 2, their corresponding poly(amic acid) (PAA) precursors (solutions) and membranes were characterized by FTIR spectroscopy.

FTIR spectra with their corresponding intensity data of the PAA precursors and PEMs are given in Appendix A, respectively. For discussion purposes, intensity data changes are shown, with emphasis given to certain areas where the success of imidization can be identified, in Figure 3 and Figure 4. 

In Figure 3, the characteristic N-H stretching vibrations of amino groups having a broad absorption peak at 3000 cm^−1^ that are found in poly(amic acids) after thermal imidization was not observed in the polyimides. The clear absorption peak at around 2882 cm^−1^ seen in Figure 3b in PEMs is due to C-H stretching of PEG, where before imidization, this aliphatic C_H stretching was also observed at around 2940 cm^−1^ for PAAs. 

Moreover, the FTIR spectrum of aromatic polyimides showed the characteristic absorptions of imide groups where the new peaks were observed at about 1780 cm^−1^ (C-O, asymmetric), 1719 cm^−1^ (C-O, symmetric), and 1392 cm^−1^ (C-N, asymmetric) respectively, as can be seen in Figure 4b. The absorption attributed to the C-F stretching vibration of trifluoromethyl groups was detected at around 1240 cm^−1^ in all spectra for PEMs [42]. The disappearance of the carbonyl stretching vibrations peak of amide groups (1635 cm^−1^) in PAA precursor can also be attributed to the thermal imidization (Figure 4a). As can be seen from Figure 3 and Figure 4, new peaks arise due to the formation of imide rings via imidization and can be assigned as the asymmetric stretching peak of the carbonyl group or the symmetric stretching peak of the carbonyl group (around 1719 cm^−1^), or the stretching vibration peak of C-N in imide ring (1392 cm^−1^), which confirms full imidization of PAA via the thermal imidization process [34,35]. Thus, it is demonstrated that the PAA precursors had been converted to the corresponding PEMs.

### 4.2. TGA Measurements

To evaluate the thermal stability of the undoped polymers, TGA measurements were conducted for all fifteen membranes. TGA results for Group 1, Group 2, and Group 3 can be seen in Figure 5, Figure 6 and Figure 7. As expected, the thermal stabilities of these membranes were found to be similar to the PEG-PI polymer systems studied previously, where they fully degraded around 600 °C but were stable up to 200 °C [28,29,30,31]. 

The tailored synthesis was a success. As can be seen from Figure 6, there is no significant weight loss before 350 °C for PEM_7–PEM_9. However, an earlier weight loss of around 300 °C was observed for PEM_10 (corresponding to ~5% mass loss up to 300 °C) and weight loss was observed at a lower temperature around 250 °C for the PEM_6 membrane (corresponding to ~10% mass loss up to 350 °C). Since PEM_7, PEM_8, and PEM_9 have both fluorinated and ether-containing dianhydrides, the end products were more stable. The second degradation at around 500 °C was observed for all the membranes. This degradation is attributed to the aromatic polyimide phase in the membranes. Since the membranes containing both types of dianhydrides showed no evidence of degradation up to 350 °C, we conclude that these materials would be suitable for the expected working conditions of the fuel cells.

In summary, the thermal degradation results are in line with the previously studied PEG-containing polyimides [28,29,30,31]. More importantly, SBC membranes containing multi-dianhydrides positively affected their properties in favor of improved thermal stability.

### 4.3. Ionic Liquid and Water Uptake

In the system design section, we reviewed and demonstrated that thermally treated membranes showed higher PEG-IL domain spacing. This structural change due to enhanced PEG phase segregation as shown by Woo et al. [31] was believed to affect the IL uptake features, where higher amounts of IL incorporation would be possible. On the other hand, to address this point for our system of 15 materials, we tested the IL uptake for both EAN and PAN as well as the uptake of water at two different temperatures: 25 and 60 °C. 

The results are summarized in Table 3, Table 4 and Table 5. Table 3 shows the results of ionic liquid uptakes (ILUs) and water uptakes (WUs) for the 6FDA + ODPA-PDODA-PEG1500 membranes in Group 1 (PEM_1 through PEM_5). Table 4 shows the results of ILUs and WUs for the 6FDA + ODPA-AP6F-PEG1500 membranes in Group 2 (PEM_6 through PEM_10). Finally, Table 5 shows the results of ILUs and WUs for the 6FDA + ODPA-PDODA + AP6F-PEG1500 membranes in Group 3 (PEM_11 through PEM_15).

In Group 1, fluorinated content was incorporated using only 6FDA and gradually decreased to incorporate oxygenated dianhydride ODPA; in more detail, PEM_1 having only 6FDA, PEM_2, PEM_3, and PEM_4 having 6FDA and ODPA content mixture in the membrane, and PEM_5 having only ODPA (all the weight percentages for the materials can be found in Table 2).

In Group 2, fluorinated content was incorporated using both 6FDA dianhydride (gradually decreased among the group) and AP6F diamine (constant concentration). 

In Group 3, fluorinated content was incorporated using both 6FDA (gradually decreased among the group) and AP6F (concentration was constant but lesser than the content in Group 2 membranes).

As a general result, when the dianhydride mixture shifted to the more ether-based one in all groups, the ILU values tended to drop for all groups. For example, EAN uptake values for the SBC membrane containing only ODPA (PEM_5) dropped by around 25% when compared with PEM_1 containing only 6FDA at 25 °C and by around 39% at 60 °C. These findings are also consistent with the structural SAXS observations of Coletta et al. that 6FDA-containing SBC membranes had more free volume, resulting in higher ILUs than the SBC membranes containing ODPA [28]. 

The same behavior was also observed for PAN uptake values in all materials. As seen in Table 3, Table 4 and Table 5, when the dianhydride mixture was changed to one containing more oxygenated dianhydride, the PAN uptake values dropped in all cases. For example, for PEM_5 membrane, PAN uptake values when compared to PEM_1 dropped by around 31% and 40% respectively, at 25 and 60 °C. 

We also wished to further explore the effect of increased fluorinated content in the SBC membrane. Thus, in Group 2, fluorinated diamine AP6F was substituted for the oxygenated diamine PDODA in Group 1. When ILU values given in Table 3 are compared with the values in Table 4, it can be seen that the ILU values increase when fluorinated diamine (AP6F) is being incorporated into the SBC membranes instead of oxygenated diamine (PDODA). For example, when PEM_1 is compared with PEM_6, EAN uptake values for PEM_6 increased by 52% at 25 °C and by 15% at 60 °C. When PEM_5 is compared with PEM_10, EAN uptake values for PEM_10 are increased by 28% at 25 °C and by 8% at 60 °C. 

Also, when ILU results (Table 5) for Group 3 materials (obtained using both AP6F and PDODA) are compared with the results given in Table 3 and Table 4, it is seen that ILU values are found to be in between. 

All these results clearly show that when fluorinated content within the material is increased either from the diamine incorporation or dianhydride incorporation, the ILU capabilities increase. These findings as explained above prove higher free volume generation in the material via the generation of well-defined PEG domains, as presented by the SAXS plots by Coletta et al. [28,29,30].

On the other hand, only for Group 1 material, WU values decreased by decreasing fluorinated dianhydride. This can be explained by the hydrophobicity of the fluorine. However, when assessing the results of Groups 2 and 3, the same WU behavior is not observed. For these groups, the WU values increased when compared to Group 1 values. Also, within each group, WU values decrease as the content of 6FDA decreases. We believe swelling of the membranes in Groups 2 and 3 is another result of well-defined PEG-domains creating free volume by the incorporation of fluorinated AP6F. Thus, more WU is realized, causing swelling within the materials. More discussion on WU is given below.

### 4.4. Effect of Increasing Temperature

When looking at PEM_1 to PEM_5, IL-doped membranes at two temperatures, it is observed that the IL uptake and WU uptake values increase with increasing temperature. These results are again in line with the findings of Woo et al. [31], where it was noted that heating caused structural change towards more well-defined phase separation where the EAN-PEG domains had wider and more distinctive SAXS peaks. Thus, we conclude that the EAN-PEG phase separation becomes dominant as the temperature increases, thus permitting more IL to be incorporated into the system. However, it should be noted that the IL uptake, especially at the elevated temperature, showed higher deviations for ionic liquid doping. These findings may reflect structural changes at the nanometer level in favor of creating a more mobile environment. In particular, the higher mobility might lead to the formation of more well-defined free volume microchannels. However, further investigation is needed.

By contrast, WU was much smaller, and the same high-level deviations were not observed. Most interestingly, WU values increased with decreasing amount of 6FDA incorporation within the SBC membrane. This can be explained through the hydrophobicity of the fluorine groups in the system. Thus, when less fluorinated dianhydride is present, higher WU values are realized. Also, at the higher temperature, uptake values were not dramatically changed when compared to ILU values. 

Another important observation is that the membranes studied at 25 °C had better strength at the end of the 24 h doping process than the membranes which were analyzed at 60 °C. This is due to the fact that at an elevated temperature, more IL could be incorporated in all the membranes. This can be seen clearly in Table 3. Thus, with the higher amount of ILU at the elevated temperature, some membranes had swollen ratios above 100%. These higher IL intakes adversely affected their mechanical strength, and more swollen membranes became weaker and tore when only a small amount of force was applied. Despite these observations, membranes could still be handled, and conductivities could be measured. However, PEMs containing a higher ratio of fluorinated dianhydride (PEM_1 and PEM_2) became much softer and were more easily torn. Thus, more care had to be given to these membranes during handling compared to PEM_3, PEM_4, and PEM_5 in order to measure the conductivities. This is further evidence that oxygenated dianhydride (ODPA) provides more mechanical strength to the membrane matrix. 

Table 4 shows the result of the SBC membranes containing fluorinated-based diamines—AP6F. Here, the same behavior was observed: with decreasing ratio of fluorinated dianhydride (6FDA) in the SBC membrane, the matrix showed decreasing ILU uptake. Also, increasing the temperature had a positive effect on ILU values, where more ILU was observed in all cases.

EAN uptake values increase around 55% for PEM_6, 34% for PEM_7, 28% for PEM_8, 11% for PEM_9, and 20% for PEM_10 when the diamine changed from PDODA to AP6F, as compared with Group 1 SBC membrane families at 25 °C. In contrast, changing the diamines had a smaller effect on EAN uptake values (increased around 7% ± 5%) when Groups 1 and 2 SBC membranes were compared for the elevated temperature. This behavior was also repeated for the PAN-doped membranes, where ILU values increased higher with changing diamine incorporation at a lower temperature than the elevated temperature.

In contrast, WU behavior surprisingly changed for these Group 2 and Group 3 SBC membranes, where decreases in the amount of fluorinated dianhydride (6FDA) were accompanied by decreases in WU values for both 25 and 60 °C. All these behavioral changes in the uptake properties, e.g., when the diamine content of the material was changed from PDODA to AP6F in Group 2, may be due to the fluorinated diamine providing more free volume in the structure than the oxygenated diamine causing more PEG domains and leading to higher free volumes. These findings are in line with what Coletta et al. observed in the SAXS plot of SBC membranes containing AP6F material [30]. The effects of changing diamines from PDODA to AP6F can be seen by the structural behavior transition from the aromatic phase to the PEG phase, where mobility increases within the structure, creating more free volume and increasing the possibility of higher intercalation of IL and water. Similar observations were reported by Lazareva et al., Low et al., and Okamoto et al. [87,88,89,90]. Lazareva et al. pointed out that more rigid fragments of chains in the structure of SBC membranes may produce domains with short-range order, where it may be supposed that such an ordering of chains renders a narrower size distribution of free volume [87]. Low et al. showed that structural modification can be obtained using diamine incorporation into the SBC membranes enabling greater free volume [88]. The same conclusions were also reported by Okamoto et al. [89,90].

In summary, ILU differences between Groups 1 and 2 are consistent with the previous studies presented by Coletta et al., where an uptake value was reported for PEM_5 and PEM_10 as 65% and 149% respectively [30], at 25 °C. As can be seen in Table 4, the value measured for PEM_15 is between the values of PEM_5 and PEM_10 as having an uptake value of 91% at 25 °C for EAN uptake.

As can be seen from Table 5, all the ILU and WU uptake values measured for Group 3 at 25 and 60 °C were found between uptake values of Group 1 and Group 2 membranes. This was expected since Group 1 materials were synthesized using only PDODA (oxygenated diamine) and Group 2 materials using only AP6F (fluorinated diamine), whereas in Group 3, both PDODA and AP6F were incorporated. This may be the consequence of incorporating oxygenated diamines where aromatic interactions caused more rigidity in the membranes and less chain swelling, which decreased the amount of intercalation of IL and water [28,87]. All these uptake values are given in Table 5, and when compared with results given in Table 3 and Table 4, show evidence of controlling ILU and WU by varying the composition of the SBC membranes. These findings suggest that controlling the composition of the different diamine allows us to enhance and fine-tune the morphology of the SBC membranes. 

Another positive observation for Group 3 family membranes is that after the doping process, these membranes were not as swollen as Group 1 and Group 2 membranes. In particular, the membranes in Group 3 maintain better mechanical strength at the elevated temperature than the rest of the materials synthesized. They were not easily torn when small stress was applied. Thus, PEM_11 through PEM_15 (Group 3 SBC membranes) showed better properties in two perspectives: swelling and mechanical strength. The integration of compositional design generates a synergistic effect, where the morphology can be fine-tuned to create SBC membranes with superior mechanical properties. Thus, understanding the effectiveness of diamines in the composition is very crucial.

### 4.5. Summary of Ionic Liquid Uptake Results 

In summary, either increasing the PEG ratio in the membrane structure increases ionic liquid uptake (ILU), as presented in References [28,29,30,31], or incorporating fluorinated diamine (AP6F) increased the ILU values. A comparison between the results acquired in this study with the results presented in References [28,29,30,31] is given below in Table 6.

### 4.6. Proton Conductivity

Conductivity measurements were carried out for all the membranes (Table 2). The tests were conducted for thermally untreated undoped membranes as well as membranes doped in EAN and PAN at ambient temperature for 24 h. Then, all these conductivity tests were repeated for 1 h-thermally annealed undoped membranes as well as membranes doped in EAN and PAN. 

Here, a detailed analysis of the impedance spectra of these major groups was carried out. The behavioral change of the impedance spectra of the membranes being untreated or thermally treated, undoped, or doped in ILs are presented in the Appendix A. 

To measure resistance from a Nyquist plot using the electrochemical impedance spectroscopy (EIS) method, there are mainly three alternatives [91,92,93,94,95,96,97]: (1) linear extrapolation of the measured spectra data to the Z’-axis of the Nyquist plot, where the Z’ value at the cross-section is taken as the membrane resistance, (2) reading Z’_real_ corresponding to the lowest value of Z’’_imaginary_ at high frequency from the impedance spectra obtained from alternating current (AC) impedance, where the semicircle starts, read is taken as the membrane resistance, or (3) applying equivalent circuit fitting.

Many equivalent circuit models have been described in the literature [91,98,99,100,101,102,103,104] but discussion of their fitting is beyond the scope of this study. However, membrane resistivity values obtained directly from Nyquist plots are also compared with the equivalent circuit models used to fit the impedance spectra of undoped and doped membranes. The equivalent circuit models used in this study to fit the impedance spectra of undoped and doped membranes are shown in Figure 8a,b, respectively. 

Since the alternating current (AC) impedance method is used, to measure the membrane resistance, the complex impedance of the sample as a function of AC frequency, ω, i.e., Z(ω) = V(ω)/I(ω), is obtained. The membrane resistance is characterized by the real component of the impedance (Z(ω) = Z’(ω)+iZ’’(ω)) at high frequencies for doped membranes and low frequencies for undoped membranes, where Z’(ω) is the real impedance and Z’’(ω) is the imaginary impedance. Thus, the real component of the impedance read from the Nyquist plots replaces R in Equation (1) for calculation of membrane conductivity. All respective values of the resistivity obtained from the two methods used in the study are given in the Appendix A. 

For all 15 membranes, the conductivities are measured for the following six different conditions:Unannealed, undoped SBC membranesUnannealed, EAN-doped SBC membranesUnannealed, PAN-doped SBC membranes1 h thermally annealed undoped SBC membranes1 h thermally annealed EAN-doped SBC membranes1 h thermally annealed PAN-doped SBC membranes

Since there is extensive data on conductivity, the values are given in tabulated format in the Appendix A. The readers are advised to access Appendix A.

When the results of Appendix A are closely assessed, there are several outcomes. To visually show these outcomes, bar charts were presented for the selected conductivity values.

Figure 9a–c, Figure 10a–c, Figure 11a–c show the conductivities for the six different conditions for Group 1, Group 2, and Group 3, respectively. In Figure 9a–c, Group 1 conductivities are shown for the six different conditions, and discussions of these observations are given in the next sections below.

#### 4.6.1. Effect of Changing Ionic Liquid

As can be seen in Figure 9, Figure 10 and Figure 11, doping the SBC samples in ILs increased conductivities of the unannealed or annealed SBC membranes. Whether or not the samples were annealed, changing the ionic liquid from EAN to PAN also had a positive effect, resulting in a higher increase of the conductivities for all the samples. There are two reasons for this increase. The first is due to having higher ionic liquid uptake abilities of the membranes when doped in PAN in almost all cases. This ability is shown via the ILU values presented in Table 3, Table 4 and Table 5. Second, this increase in conductivity can be due to the additional alkyl-chain CH_2_ in PAN as compared to EAN, which can generate an environment where more well-defined segregation of the polar and non-polar regions are obtained, thus enhancing the conductivity.

#### 4.6.2. Effect of Thermal Annealing 

Woo et al.’s [31] SAXS results showed structural changes occurring with increasing temperature. It is believed that through this thermal annealing process, the morphologies of the membranes change, creating more well-defined PEG-IL microchannels. These microchannels play a crucial role in easing the ion transfer causing the conductivity to increase. More significantly, once the membranes are thermally annealed and a structural change has taken place at the molecular level, these changes were still present upon cooling [31].

To understand the effect of thermal annealing on the PI-PEG membranes, a more comprehensive study was carried out for our 15 samples using six different conditions, as mentioned above. When conductivities of undoped membranes at lower and higher temperatures are compared (Figure 9a, Figure 10a, and Figure 11a), significant increases in all materials are observed at the higher temperature. To emphasize this impact, the results of PEM_14 were used in the following part. Also, to clarify the reason for the increase in their conductivities, impedance spectra of undoped membranes at 25 and 60 °C are also given in Appendix A. When the spectra of these two conditions are compared, the values of real impedances, or more specifically, their conductivities, change dramatically; for example, for PEM14, the conductivity increased from 2.65 × 10^−3^ to 5.50 × 10^−2^ mScm^−1^ after thermal annealing (Appendix A or Figure 11a). This positive effect of thermal annealing on conductivities for all undoped membranes was found to be the same as PEM_14.

Another observation seen during the experiments is that the conductivity measurements were repeated three times for all SBC membranes. Thus, time was passing during the measurements, and measurements of the conductivities were conducted under room temperature, where the thin films were quickly cooling down to room temperature. However, the outcome of the measurements of the second and third membrane films did not deviate from the first measurements. It can be supposed that the structural improvement provided by the thermal annealing is not temporary, which is in line with what Woo et al. observed in the SAXS plot upon cooling of the films [31]. 

In addition, when the impedance spectra of the doped membranes at 25 and 60 °C are compared (Appendix A), there are several behavioral differences. The membranes for which the doping process took place at the lower temperature had a wider first half-circle, which may be attributed to the morphology of the surface. The impedance spectra obtained for the samples at the elevated temperature showed a different equivalent electrical circuit. The change in the morphology (and the change in the ion transport capability) due to the irreversible annealing can be clearly seen from these spectra and from their conductivity data (Figure 9, Figure 10 and Figure 11). This may be explained by Woo et al.’s SAXS results given for increasing temperature, where they showed that after thermal annealing, the EAN-PEG domains became more ordered [31]. The structural differences were believed to be related to the variation of the interfacial roughness. These SAXS observations also align with the impedance spectra change seen in this study. To elaborate on this point, in Figure 12, conductivity results for the PEM_1 membrane are compared with Woo et al.’s and Coletta et. al.’s findings [28,31].

As can be seen in Figure 12, Coletta et al. studied the 50% PEG-containing membrane but did not do any thermal annealing. By contrast, Woo et al. studied the 42% PEG-containing membranes for both unannealed and thermally annealed membranes [31]. Comparing the unannealed results with our findings, Coletta et al.’s results are superior. This is because higher amounts of PEG were incorporated, which would be expected to lead to the formation of more well-defined microchannels, thus promoting ion transfer, but at the cost of poorer mechanical properties. Thus, lower rigidity and lower strength were observed in the higher PEG-containing SBC membranes. On the other hand, Woo et al.’s conductivity result for the unannealed membrane was found to be very close with PEM_1: 62 and 75.56 mScm^−1^, respectively [31]. As can be seen in Figure 12, thermal annealing significantly increased conductivity data by a factor of 3–4 in both cases. 

In the analysis of the Nyquist plots (given in the Appendix A, and Figure 12) for IL doped membranes at a lower temperature, it was observed that the straight line had a smaller slope than the one thermally annealed at 60 °C with respect to the real axis. This may also indicate that the morphology of the membranes changed dramatically. This observation seen in the Nyquist plot is in line with Woo et al., where interfacial roughness was found to dominate over EAN-PEG or PAN-PEG domains, affecting both solution resistance in the membrane and charge transfer resistance [31]. Also, when the resistivity values of the doped membranes fell below 40 ohms, the membranes were observed to be very soft, very swollen, and hard to handle. However, most of the synthesized materials were found to have good mechanical strength, and especially Group 3 membranes were superior to Groups 2 and 1.

As can be seen in Figure 9 and Figure 10, the conductivity levels for Group 1 families are higher than Group 2 families in almost all cases. The reader is also encouraged to look at Appendix A. These findings also align with our ILU results. Also, when the conductivity values of Group 3 were compared with Groups 1 and 2, they were found to be closer to the values of Group 1 membranes. Most importantly, Group 3 possessed superior mechanical properties compared to the other two groups. 

In summary, the data show a significant increase in conductivity for the doped membranes at 25 °C and even higher conductivity values for thermally annealed membranes at 60 °C.

#### 4.6.3. Effect of Changing Diamines 

When diamines were changed from oxygenated to fluorinated components, the conductivities were lower (see the differences between Group 1 (Figure 9a–c) and Group 2 (Figure 10a–c)). This confirms the structural changes presented by Coletta et al., where they interpreted that systems containing fluorinated diamine-containing families (ODPA-AP6F-PEG and 6FDA-AP6F-PEG, both having 50 wt% of PEG content) had more free volume. Further, Coletta et al. reported much lower conductivity for ODPA-PDODA-PEG-containing SBC membrane and correlated this result with SAXS findings, where a more rigid and more aromatic ordered domain with lesser free volume structure was observed. Thus, Coletta et al. used SAXS experiments to obtain an explanation for the differences in ILU values affecting conductivity. It was reported that lower free volume in the structure led to lower ILU values and the conductivities of the membranes were not greatly enhanced [28]. This result is also present in our specific membranes: PEM_5 and PEM_10.

One can create more well-defined IL-PEG segments, allowing more interconnected channels for ion transport. Woo et al. studied the effect of changing PEG concentrations in the 6FDA-PDODA-PEG1500 family in order to observe the structural changes as well as the conductivity values [31]. Woo et al. reported that the highest conductivity was measured for the 6FDA-PDODA-PEG1500 membrane with 42.1 wt% PEG1500, as 210 mScm^−1^ at 60 °C doped in EAN [31]. By contrast, in the present study, the highest conductivity was found for PEM_1 as 396.61 mScm^−1^ at 60 °C doped in PAN.

In addition, Group 3 membrane families had higher conductivity than Group 2. This increase is believed to be due to the membranes having more defined IL-PEG domains, which strongly affects the transport properties. 

#### 4.6.4. Effect of Changing Dianhydrides 

In order to better understand this effect, PEM_1, PEM_5, PEM_6, PEM_10, PEM_11, and PEM_15 should be compared. Samples PEM_1, PEM_6, and PEM_11 only contain fluorinated dianhydride; by contrast, PEM_5, PEM_10, and PEM_15 consist of only ether-containing dianhydride. When these membranes are compared, it can be seen that membranes with fluorinated dianhydride showed higher conductivities in most of the cases. 

When untreated membranes are compared, it can be seen that PEM_11 and PEM_15 had higher conductivities than the other membranes (PEM_1, PEM_5, PEM_6, and PEM_10). However, a more dramatic result can be seen for the PAN-doped PEM_11 and PEM_15, where changing the dianhydride to ODPA material resulted in a conductivity reduction by a factor of 3. 

When undoped thermally annealed and not thermally treated membranes are compared, Groups 1 and 3 membranes showed higher values than the Group 2 membranes. 

In summary, thermally annealed membranes have been shown to have higher conductivities, presumably due to nanometer-scale structural change, where this effect was maintained upon cooling. 

### 4.7. Tensile Strength 

Different polyimide structures were designed to provide strength, flexibility, and conductivity in a single SBC membrane. Membranes in a confined environment should withstand dimensional swelling and nonlinear deformation stresses. In order to show the mechanical performances of the synthesized SBC membranes and quantitatively show the results of thermal annealing, mechanical tests were conducted. Woo et al. published mechanical properties of 6FDA-PDODA-PEG1500 SBC membranes for changing PEG1500 concentration for the undoped condition at ambient temperature [31]. It was shown that with increasing PEG content in the SBC membrane matrix, the tensile modulus decreased. Tensile testing at elevated temperatures is also very informative; thus, the SBC membranes under study for this work were tested for the following two conditions:SBC undoped membranes without thermal treatmentSBC undoped membranes with thermal treatment at 60 °C

A typical stress-strain curve is shown in Figure 13 for group 2 (PEM_6–PEM_10). Also, in the same figure, Nafion 115 is shown for comparison purposes.

As expected, when 6FDA incorporation gradually decreased from PEM_6 to PEM_10, a clear transition existed from ductile to brittle behavior. From Figure 13, one can see that Nafion 115 showed increasing stress as it was strained past the yield point, whereas PEM_6 through PEM_9 did not show this behavior. In contrast, only PEM_10 showed a positive slope after the yield point, like Nafion 115, which is an indication of strain hardening. This is further evidence that the mechanical strength of the membranes is increased by the incorporation of the oxygen-containing dianhydride ODPA. This confirms the observations seen in thermal annealing, where the membranes with higher fluorinated base dianhydride content became softer. 

In Table 7, the tensile stress at maximum force and Young’s modulus are given for every synthesized SBC membrane. These values show how the structural changes affect the mechanical strengths. All undoped and thermally untreated SBC membranes exhibit tensile stress at maximum force around 10 MPa (having a higher 6FDA amount in the polyimide structure) or higher at ambient temperature. As expected, tensile stress values decreased by around half for all the membranes when they were thermally annealed. However, to define their mechanical properties, Young’s modulus becomes the best metric. 

An ideal Young’s modulus value can occur over the broad range of ~65–450 MPa, where handling was found to be easier according to Ertem et al., no fluid-like behavior was observed, and ruptures due to physical contact were not observed [105]. Below 65 MPa, the membrane can be more fluid-like, and above 450 MPa, the membrane can be very stiff [106,107]. Another important requirement is to keep Young’s Modulus difference to a minimum at 25 and 60 °C. Nafion 115 meets these requirements with the values of 288.94 and 221.46 MPa at 25 and 60 °C, respectively. 

When PEM_1 through PEM_5 membranes are analyzed, the change in the Young’s modulus was high and the values were even below 65 MPa for PEM_1 and PEM_2, indicating they became very soft under the higher temperature. But PEM_3 through PEM_5 had Young’s modulus above 75 MPa, which is suitable for use in a confined environment. Also, it should be kept in mind that the membranes are not under extensional tension but are mainly subject to oscillations in force, temperature, and relative humidity in a working electrochemical device. These oscillations occurring in the device are not strong enough to destroy the membrane, such that very high strengths are not required. As long as a membrane maintains its mechanical integrity during conductivity testing, even though presenting a lower Young Modulus, it may still be a viable option for practical devices [104]. 

When PEM_6 through PEM_10 membranes were analyzed, incorporation of AP6F increased the elasticity of the membranes. This change can be observed from the decreases seen in their tensile stress at maximum force and Young’s modulus values at ambient temperature. As expected, membranes with a higher amount of 6FDA (PEM_6 and PEM_7) showed a higher decrease in their Young’s modulus at elevated temperature than PEM_8, PEM_9, and PEM_10, where ODPA incorporation increases. 

Finally, when PEM_11 through PEM_15 membranes were analyzed, it can be seen that incorporation of different functional dianhydrides and diamines had a positive effect on the mechanical properties in those deviations of Young’s modulus, which were lowered with increasing temperature. The tensile test data also confirms the observation seen in the thermal annealing process, where all of the membranes in Group 3 were not softened, as opposed to Group 1 and Group 2 membranes, where a higher amount of 6FDA-incorporated membranes were softened. This behavior can be easily seen in Figure 14.

In the literature, it is stated that for an undoped membrane, elongation values should be greater than 100% as a conservative metric [104]. In fact, all of the undoped and thermally untreated membranes exhibit elongation higher than 100%, except PEM_3, PEM_4, PEM_5, PEM_10, and PEM_15, where their elongation values were found to be lower than 50%. This decrease in elongation is believed to be the result of higher ODPA content in the membrane structure where it gives more rigidity. Rigidity is linked to being harder, stiffer, and less flexible, and this morphology was also interpreted by Coletta et al. via SAXS plots, where more well-defined ordered aromatic polyimide structures were observed. Thus, our findings from tensile experiments also confirm that the oxygen-containing SBC membrane should possess lower elongation. 

## 5. Conclusions

An extensive study was carried out in order to have more insight on the relationship of molecular structure to properties for a PI-PEG SBC membrane system using dianhydrides and diamines with different functionalities. It has been shown that the morphology was strongly influenced by thermal annealing, which led to increases in ILU and conductivities. These findings were consistent with both Woo et al. and Coletta et al., where they suggested that ion channels were being created in the structure via thermal annealing and incorporating IL into PEG domains [28,29,30,31]. Our results clearly support this idea. When switching from PDODA diamine to AP6F diamine for 42.1 wt% PEG1500, the membranes became more elastic, and higher ionic liquid uptake values were obtained due to increased IL-PEG-domain microchannels. When incorporating both diamines in the SBC membrane matrix, stiffer and more stable membranes could be synthesized. When the dianhydride composition was changed by gradually decreasing 6FDA and increasing ODPA, there was a clear transition from the ductile to the stiffer membrane. Also, incorporation of higher ODPA amounts in the membranes lowered ILU values, where the conductivities decreased. It was shown that thermal annealing increases the ionic liquid incorporation, where the conductivity significantly increased by more than a factor of 5 compared to Nafion. By varying the IL type from EAN to PAN, the conductivity was increased. The maximum conductivity was obtained for PEM_1 SBC membrane at elevated temperature and it was found to be much greater (nearly six-fold) than that of Nafion 115. From all these evaluations, we have shown that the properties can be engineered through structural and compositional changes, where required properties can be evaluated. More optimization can be realized in this set of materials in order to tune the properties required for various applications, such as gas separation, actuators, or battery electrolytes.

## Figures and Tables

**Figure 1 molecules-26-02143-f001:**
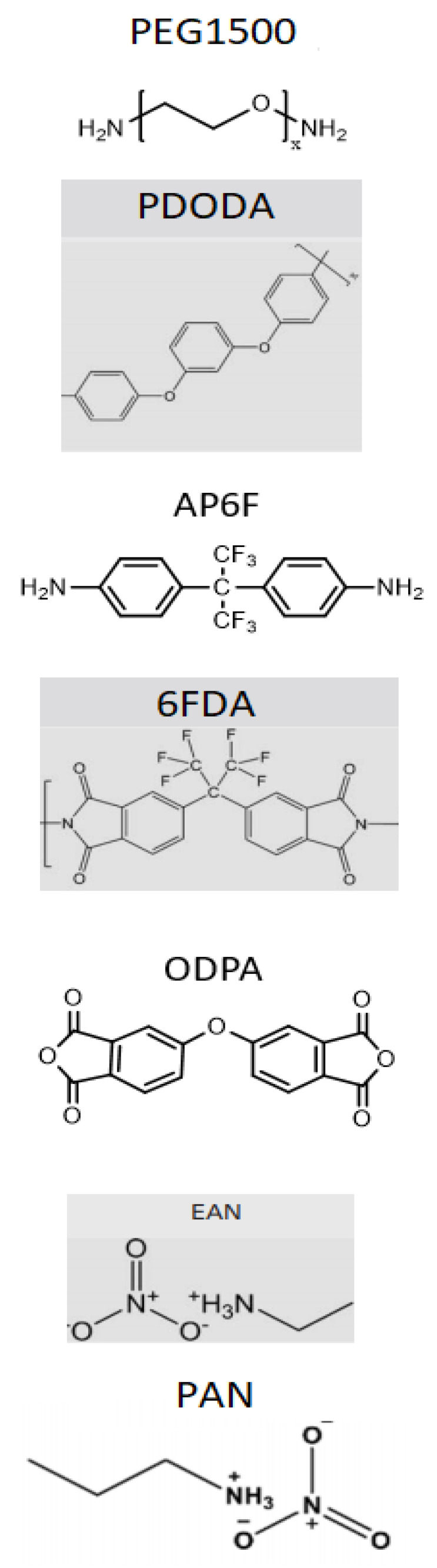
Chemical structures of the materials used in the two-step production method.

**Figure 2 molecules-26-02143-f002:**
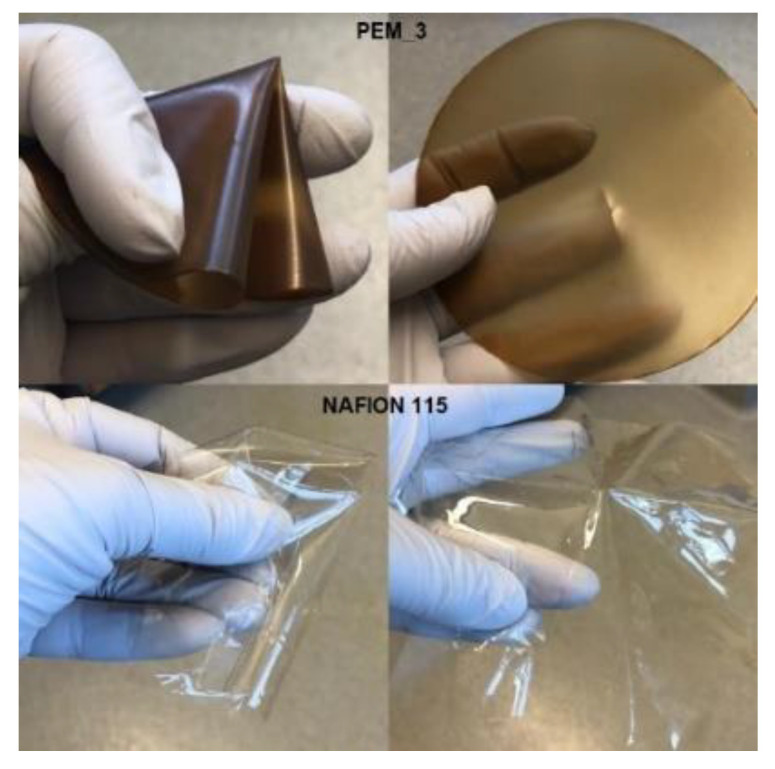
Comparing the final product with Nafion 115 using the non-mechanical folding test.

**Figure 3 molecules-26-02143-f003:**
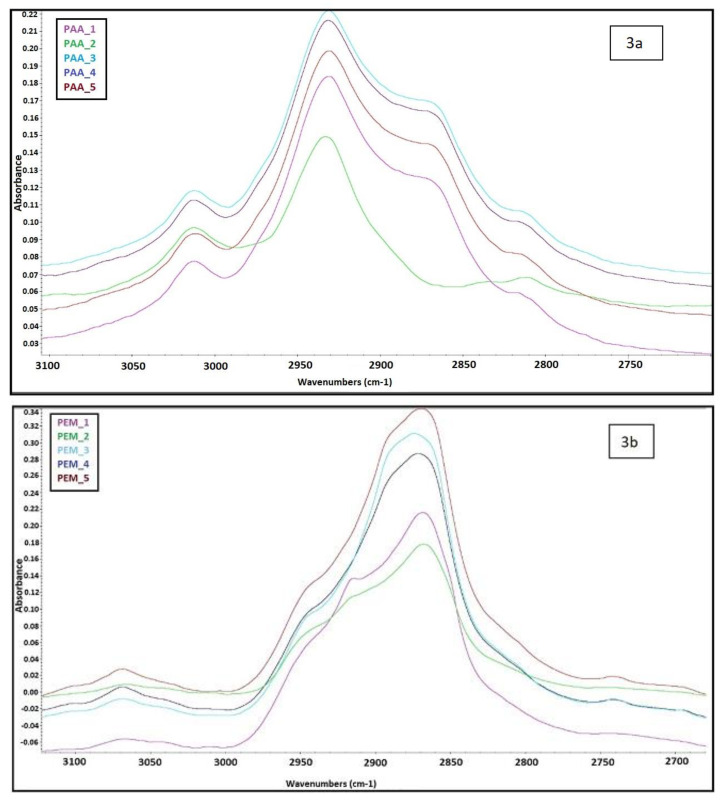
Zoomed Fourier transform infrared (FTIR) spectra of poly (amic acid)_1-poly (amic acid)_5 (PAA_1–PAA_5) solutions (**a**) and PEM_1–PEM_5 membranes (**b**) between 3100 and 2700 cm^−1.^

**Figure 4 molecules-26-02143-f004:**
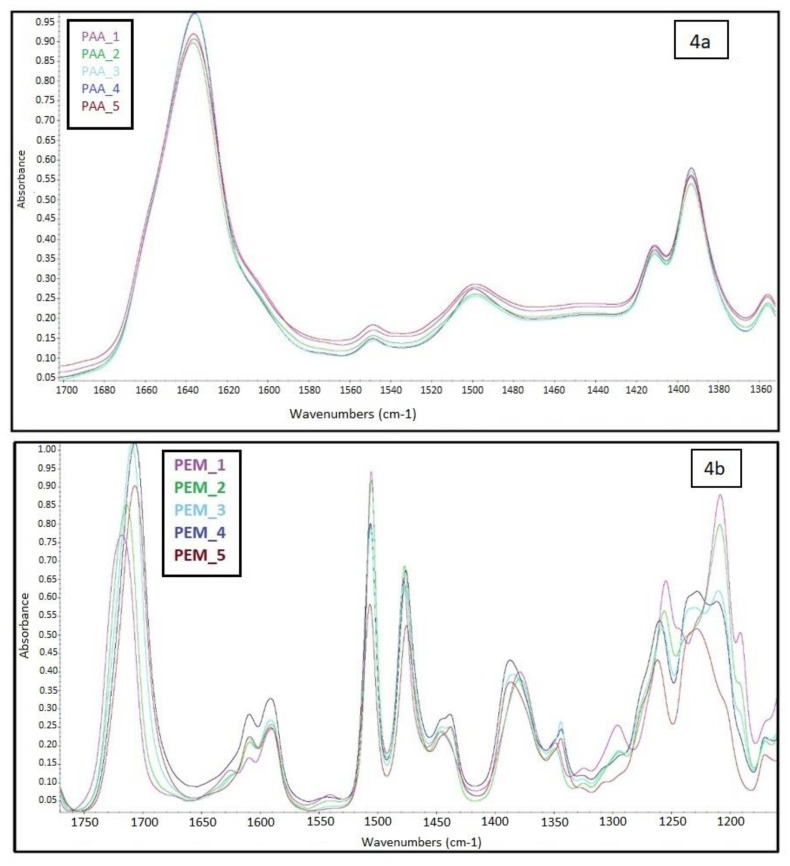
Zoomed FTIR spectra of PAA_1–PAA_5 solutions (**a**) and PEM_1–PEM_5 membranes (**b**) between 1750 and 1200 cm^−1.^

**Figure 5 molecules-26-02143-f005:**
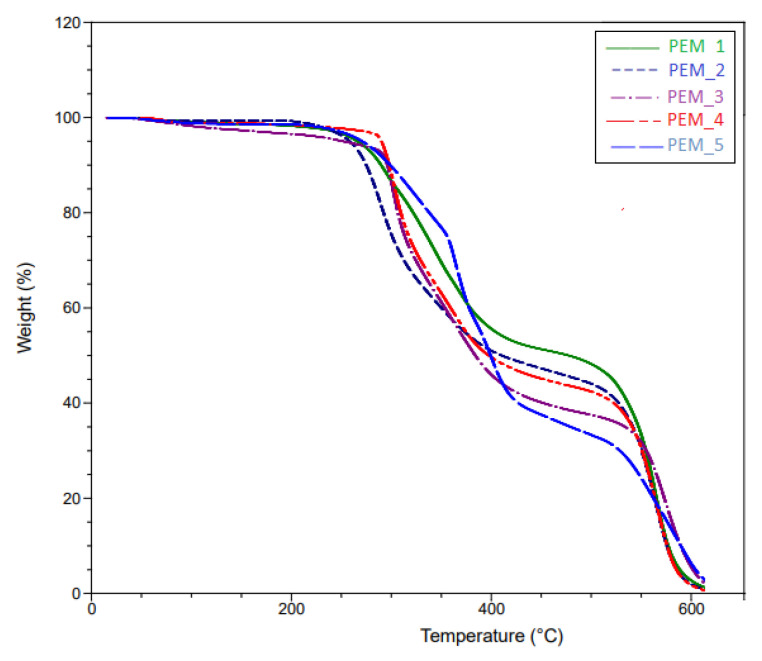
Thermal gravimetric analysis (TGA) curves of PEM_1–PEM_5.

**Figure 6 molecules-26-02143-f006:**
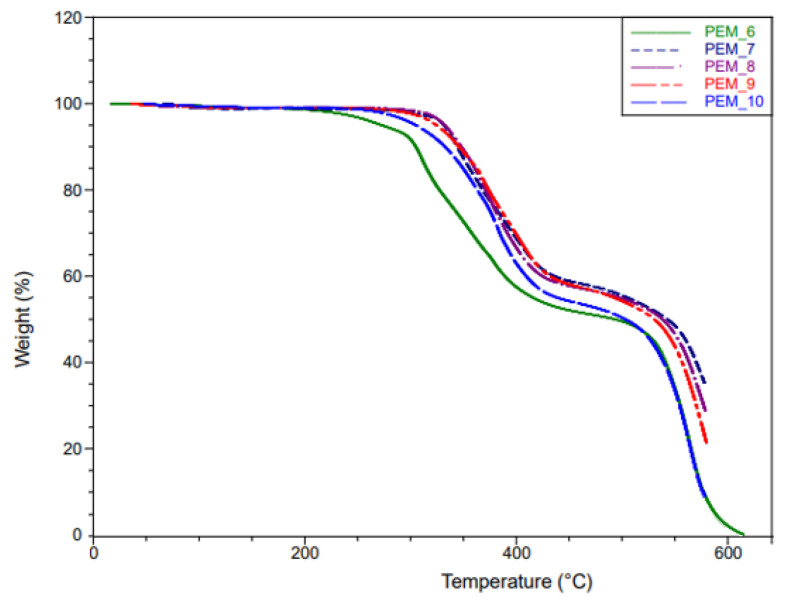
TGA curves of PEM_6–PEM_10.

**Figure 7 molecules-26-02143-f007:**
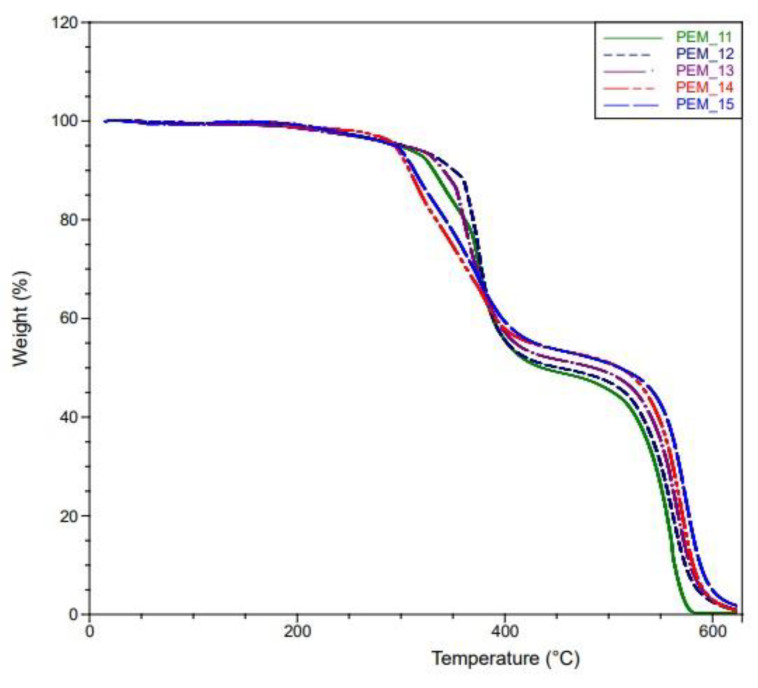
TGA curves of PEM_11–PEM_15.

**Figure 8 molecules-26-02143-f008:**
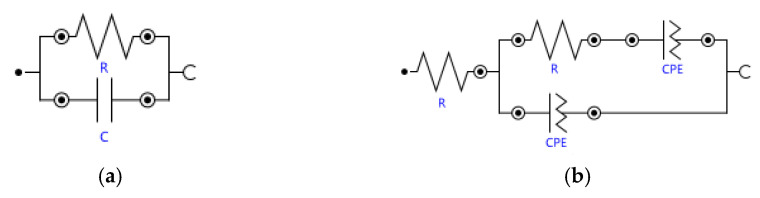
Equivalent circuit models used in this study: (**a**) impedance spectra of undoped membranes and (**b**) impedance spectra of doped membranes. In some cases, either a series inductor or Warburg impedance was included in the model to account for high-frequency inductance or diffusion artifact in the data.

**Figure 9 molecules-26-02143-f009:**
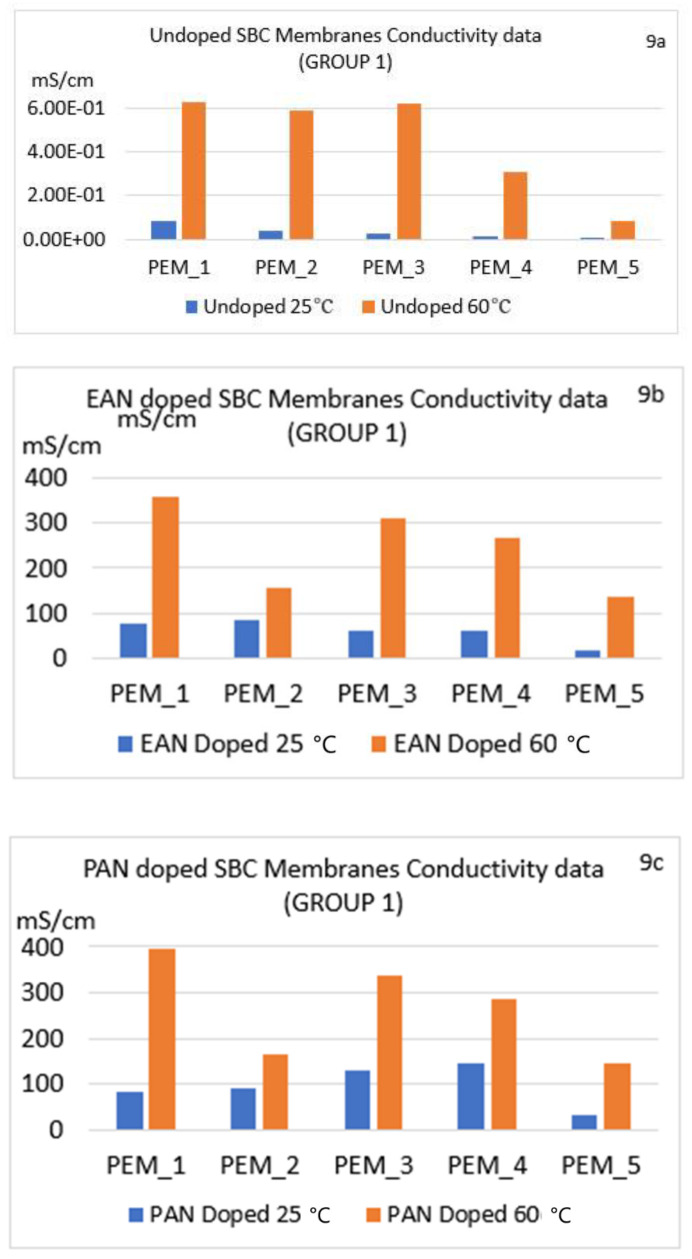
Comparison of conductivity values for nonannealed and annealed SBC membranes for all conditions for Group 1 membranes: PEM_1–PEM_5 (undoped and doped). (**a**) Undoped (**b**) EAN doped (**c**) PAN doped.

**Figure 10 molecules-26-02143-f010:**
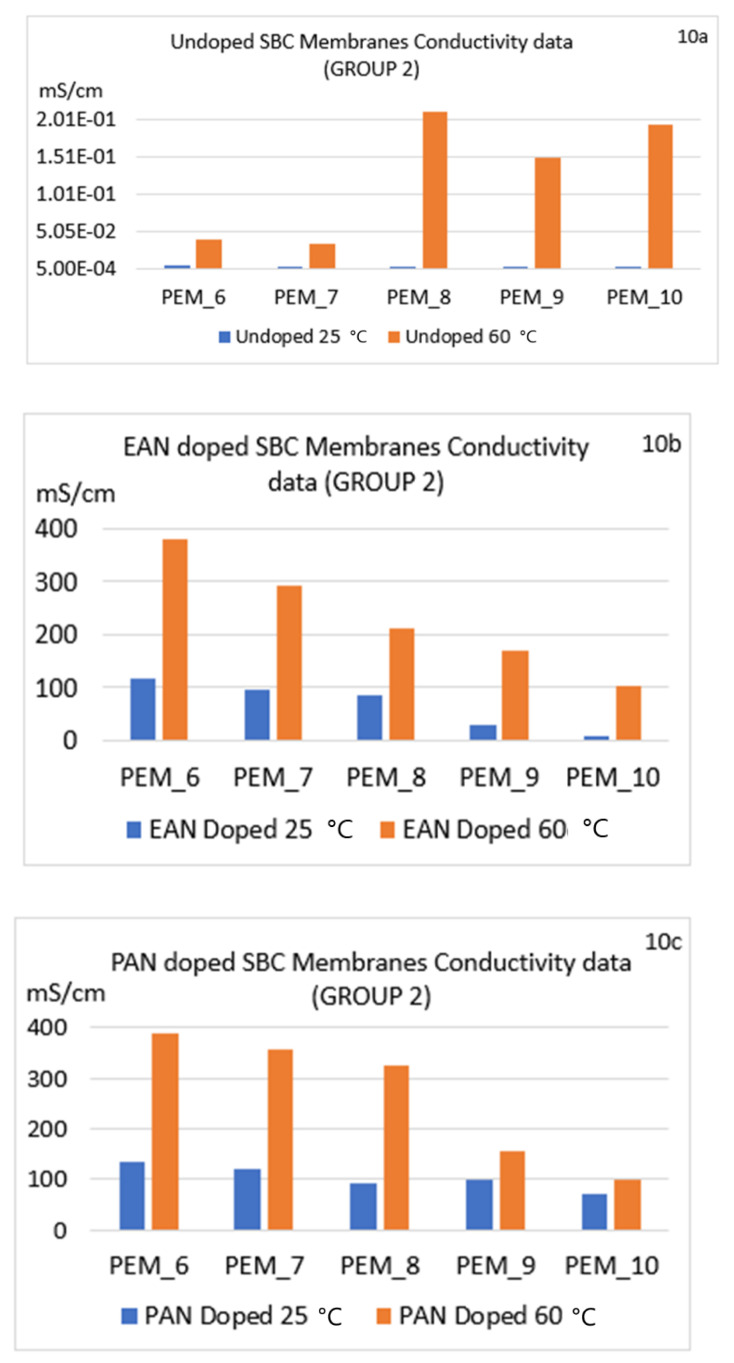
Comparison of conductivity values for nonannealed and annealed SBC membranes for all conditions for Group 2 membranes: PEM_6–PEM_10 (undoped and doped). (**a**) Undoped (**b**) EAN doped (**c**) PAN doped.

**Figure 11 molecules-26-02143-f011:**
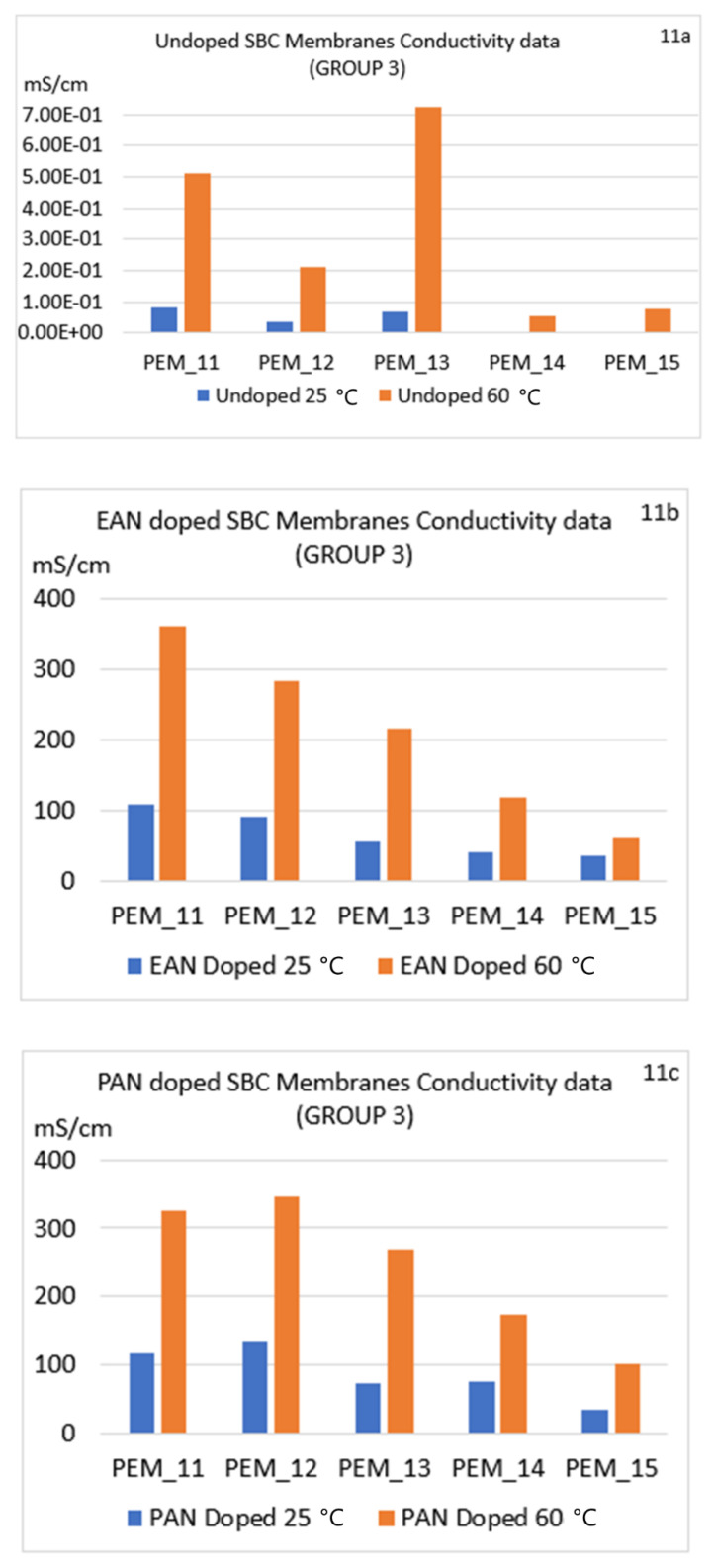
Comparison of conductivity values for nonannealed and annealed SBC membranes for all conditions for Group 3 membranes: PEM_11–PEM_15 (undoped and doped). (**a**) Undoped (**b**) EAN doped (**c**) PAN doped.

**Figure 12 molecules-26-02143-f012:**
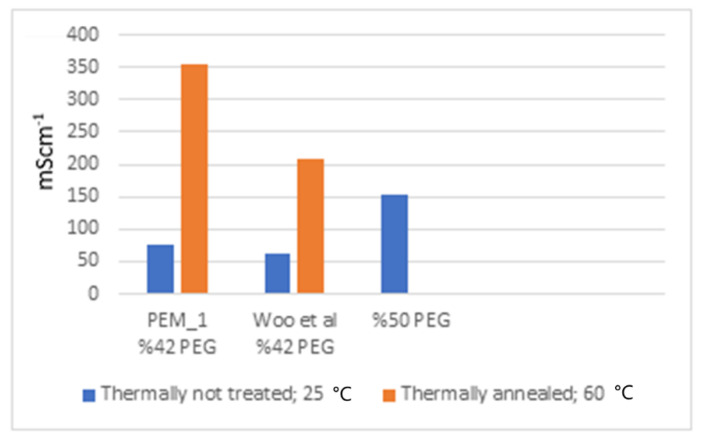
Comparison of conductivity of the EAN-doped PEG-PI membranes containing 6FDA-PDODA-PEG1500.

**Figure 13 molecules-26-02143-f013:**
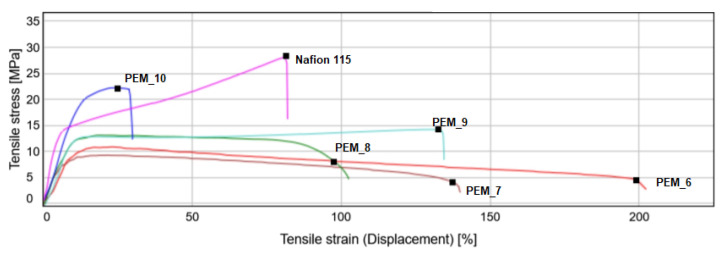
Stress-strain curve for Group 2 SBC membrane families.

**Figure 14 molecules-26-02143-f014:**
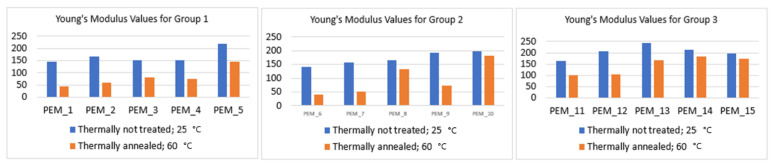
Comparison of thermally not treated and thermally annealed 15 membranes’ Young’s modulus values.

**Table 1 molecules-26-02143-t001:** Comparison of polyimide-poly(ethylene glycol) segmented block copolymer (PI-PEG SBC) membranes synthesized previously by Coletta et al. [28,29,30] and Woo et al. [31] with the membranes synthesized in this work.

Coletta et. al. [28,29,30]	ODPA-PDODA-PEG1500 (50%)	ODPA-AP6F-PEG1500 (50%)	ODPA-AP6F-PEG6000 (50%)	6FDA-PDODA-PEG1500 (50%)	6FDA-AP6F-PEG1500 (50%)	
		ODPA-AP6F-PEG1500 (40%)	ODPA-AP6F-PEG3400 (50%)			
		ODPA-AP6F-PEG1500 (30%)	ODPA-AP6F-PEG2000 (50%)			
			ODPA-AP6F-PEG990 (50%)			
Woo et. al. [31]				6FDA-PDODA-PEG1500 (46.8%)		
				6FDA-PDODA-PEG1500 (42.1%)		
				6FDA-PDODA-PEG1500 (33.6%)		
				6FDA-PDODA-PEG1500 (26.2%)		
Ciftcioglu et. al.	Group 1			Group 2		Group 3
	ODPA-PDODA-6FDA-PEG1500 (50%)			6FDA-ODPA-AP6F-PEG1500 (42.1%)		ODPA-PDODA-6FDA-AP6F-PEG1500 (42.1%)
	See Table 2 for detailed weight percentages			See Table 2 for detailed weight percentages		See Table 2 for detailed weight percentages

**Table 2 molecules-26-02143-t002:** Weight percentages of the materials used in the synthesis of the membranes.

	Materials	PEG1500	PDODA	AP6F	6FDA	ODPA
Membranes						
Proton Exchange Membrane_1 (PEM_1)		42.10 wt%	27.90 wt%	0.00 wt%	30.00 wt%	0.00 wt%
Proton Exchange Membrane_2 (PEM_2)		42.10 wt%	27.90 wt%	0.00 wt%	22.50 wt%	7.50 wt%
Proton Exchange Membrane_3 (PEM_3)		42.10 wt%	27.90 wt%	0.00 wt%	15.00 wt%	15.00 wt%
Proton Exchange Membrane_4 (PEM_4)		42.10 wt%	27.90 wt%	0.00 wt%	7.50 wt%	22.50 wt%
Proton Exchange Membrane_5 (PEM_5)		42.10 wt%	27.90 wt%	0.00 wt%	0.00 wt%	30.00 wt%
Proton Exchange Membrane_6 (PEM_6)		42.10 wt%	0.00 wt%	27.90 wt%	30.00 wt%	0.00 wt%
Proton Exchange Membrane_7 (PEM_7)		42.10 wt%	0.00 wt%	27.90 wt%	22.50 wt%	7.50 wt%
Proton Exchange Membrane_8 (PEM_8)		42.10 wt%	0.00 wt%	27.90 wt%	15.00 wt%	15.00 wt%
Proton Exchange Membrane_9 (PEM_9)		42.10 wt%	0.00 wt%	27.90 wt%	7.50 wt%	22.50 wt%
Proton Exchange Membrane_10 (PEM_10)		42.10 wt%	0.00 wt%	27.90 wt%	0.00 wt%	30.00 wt%
Proton Exchange Membrane_11 (PEM_11)		42.10 wt%	13.95 wt%	13.95 wt%	30.00 wt%	0.00 wt%
Proton Exchange Membrane_12 (PEM_12)		42.10 wt%	13.95 wt%	13.95 wt%	22.50 wt%	7.50 wt%
Proton Exchange Membrane_13 (PEM_13)		42.10 wt%	13.95 wt%	13.95 wt%	15.00 wt%	15.00 wt%
Proton Exchange Membrane_14 (PEM_14)		42.10 wt%	13.95 wt%	13.95 wt%	7.50 wt%	22.50 wt%
Proton Exchange Membrane_15 (PEM_15)		42.10 wt%	13.95 wt%	13.95 wt%	0.00 wt%	30.00 wt%

**Table 3 molecules-26-02143-t003:** Uptake values of PEM_1–PEM_5 membranes.

	Doped in EAN at 25 °C		Doped in EAN at 60 °C		Doped in PAN at 25 °C		Doped in PAN at 60 °C		Doped in WU at 25 °C		Doped in WU at 60 °C	
	Average Value of ILU%	Standard Deviation	Average Value of ILU%	Standard Deviation	Average Value of ILU%	Standard Deviation	Average Value of ILU%	Standard Deviation	Average Value of WU%	Standard Deviation	Average Value of WU%	Standard Deviation
PEM_1	106	14	195	17	148	11	242	12	22	10	41	6
PEM_2	103	11	173	18	125	10	217	14	36	5	49	2
PEM_3	89	11	154	18	119	12	190	15	42	10	53	4
PEM_4	88	9	141	17	109	12	187	15	43	8	54	6
PEM_5	80	9	135	11	95	10	146	13	45	10	58	5

**Table 4 molecules-26-02143-t004:** Uptake values of PEM_6–PEM_10 membranes.

	Doped in EAN at 25 °C		Doped in EAN at 60 °C		Doped in PAN at 25 °C		Doped in PAN at 60 °C		Doped in WU at 25 °C		Doped in WU at 60 °C	
	Average Value of ILU%	Standard Deviation	Average Value of ILU%	Standard Deviation	Average Value of ILU%	Standard Deviation	Average Value of ILU%	Standard Deviation	Average Value of WU%	Standard Deviation	Average Value of WU%	Standard Deviation
PEM_6	164	11	224	16	183	10	254	15	69	4	84	4
PEM_7	138	13	186	17	165	12	235	13	41	2	50	2
PEM_8	120	12	170	17	152	12	207	14	39	4	45	2
PEM_9	118	13	145	18	121	9	200	15	24	2	31	3
PEM_10	102	12	139	15	114	10	192	12	30	2	42	2

**Table 5 molecules-26-02143-t005:** Uptake values of PEM_11–PEM_15 membranes.

	Doped in EAN at 25 °C		Doped in EAN at 60 °C		Doped in PAN at 25 °C		Doped in PAN at 60 °C		Doped in WU at 25 °C		Doped in WU at 60 °C	
	Average Value of ILU%	Standard Deviation	Average Value of ILU%	Standard Deviation	Average Value of ILU%	Standard Deviation	Average Value of ILU%	Standard Deviation	Average Value of WU%	Standard Deviation	Average Value of WU%	Standard Deviation
PEM_11	148	10	216	15	171	12	245	14	58	4	74	5
PEM_12	123	9	181	14	146	8	225	14	42	4	49	4
PEM_13	105	12	159	14	140	12	188	15	34	5	44	3
PEM_14	101	11	142	11	115	10	180	16	33	3	39	3
PEM_15	91	12	137	9	100	10	169	16	28	4	36	2

**Table 6 molecules-26-02143-t006:** ILU comparison between the results acquired in this study with the results presented in references [28,29,30,31].

PEG-PI Systems	EAN Uptake (%) at 25 °C	EAN Uptake (%) at 60 °C
ODPA-AP6F-PEG1500 (50%) [28]	149	Not studied
ODPA-PDODA-PEG1500 (50%) [28]	65	Not studied
ODPA-AP6F-PEG1500 (40 wt%) [29]	5	Not studied
ODPA-AP6F-PEG1500 (50 wt%) [29]	149	Not studied
ODPA-AP6F-PEG1500 (50 wt%) [30]	149	Not studied
6FDA-AP6F-PEG1500 (50 wt%) [30]	203	
6FDA-PDODA-PEG1500 (42.1 wt%) [31]	62	Not studied
6FDA-PDODA-PEG1500 (42.1 wt%) (PEM_1)	106	195
ODPA-PDODA-PEG1500 (42.1 wt%) PEM_5	80	135
6FDA-AP6F-PEG1500 (42.1 wt%) (PEM_6)	164	224
ODPA-AP6F-PEG1500 (42.1 wt%) PEM_10	102	139

**Table 7 molecules-26-02143-t007:** Tensile stress at maximum force and Young’s modulus for all synthesized SBC membranes.

	Membranes Not Thermally Treated		Membranes Thermally Annealed for 1 h	
	Tensile Stress at Maximum Force (MPa)	Young’s Modulus(MPa)	Tensile Stress at Maximum Force (MPa)	Young’s Modulus(MPa)
PEM_1	9.62	147.05	6.50	45.6
PEM_2	11.67	166.92	4.03	59.54
PEM_3	18.21	150.43	6.71	79.83
PEM_4	14.53	152.18	7.24	75.30
PEM_5	17.43	220.33	10.80	145.97
PEM_6	10.88	140.21	3.75	41.29
PEM_7	9.29	158.65	3.00	52.24
PEM_8	13.12	164.94	4.56	133.02
PEM_9	14.22	193.44	3.05	72.03
PEM_10	22.23	198.44	10.12	180.58
PEM_11	11.23	165.26	5.12	101.32
PEM_12	27.82	206.78	6.83	103.57
PEM_13	14.25	243.24	7.33	166.63
PEM_14	15.76	213.67	8.31	183.31
PEM_15	18.65	195.35	11.77	175.07

## Data Availability

Not applicable.

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
