# Peer review of "Effect of Increased Ionic Liquid Uptake via Thermal Annealing on Mechanical Properties of Polyimide-Poly(ethylene glycol) Segmented Block Copolymer Membranes"

_molecules, 2021, doi:10.3390/molecules26082143_

Round 1

Reviewer 1 Report

Ciftcioglu et al reported by “Effect of Increased Ionic Liquid Uptake via Thermal Annealing on Mechanical Properties of Polyimide-Poly(ethylene glycol) Segmented Block Copolymer Membranes”. I suggested comments below:

  1. The overall manuscript has a lot of typo errors or mistakes. should you check carefully the manuscript?
  2. Figures 2 and 3 qualities are poor, should be improved.
  3. What about PAA-1 to PAA-5 TGA data compare with a PEM-6 to 10. Figure 4 shows the PEM-6 membrane is more stable after 780 °C but the other membrane is not stable, Why?
  4. Figure 6 quality is poor, I didn’t see clearly, and PAA-1 to PAA-5 and PEM membrane results compare to explain.
  5. What about PAA membrane data of Stress-strain.
  6. Why the PAA membrane prepared didn’t compare with PEM membrane data. What is the role of PAA membrane preparation?   

Reviewer 2 Report

This is an extensive work about performances of doped and undoped block copolymer membranes. There are no new structural characterizations, but results are commented based on same or close systems reported in literature. Due to the large amount of new results, this work will certainly merit publication. Due to the large amount of new results, this work will certainly merit publication. Authors have carefully analyzed fine variations in the sample series, but there are no global scientific discussion and no clear summary of outcomes. I think that this manuscript cannot be accepted in its present form, but I warmly recommend resubmission in Molecules, by considering the following points:

  1. Readership of Molecules is mainly constituted by chemists and physical chemists. So, it would be of great help for the future readers to have a scheme listing all chemical structures, as for instance done in reference [31].
  2. Since many comments are based on comparisons with results of references [28-31], authors should specify relations between previous systems and new ones listed in table 1.
  3. After presenting and discussing their individual results, authors should summarize their findings and their comparisons with references [28-31].
  4. This is only a recommendation: impact of future article might be higher if a part of the results were transferred to ESI.

Round 2

Reviewer 1 Report

The authors are response the review comments, I suggested to accepted the revised manuscript in Molecules Journal. 

Reviewer 2 Report

Authors have completed their manuscript with missing information and comments. I recommend acceptation of the manuscript after consideration of following minor points:

  1. Figure 1 was added by authors and gives an overview of the chemical structures, which will be of great help for future readers. However, the image resolution should be increased for a better readability of small titles. Also the formula of AP6F is missing. Furthermore, I suggest to add EAN and PAN to have a complete correspondence table between references and formulas.

  1. The image resolution of the graphs in figure 9 should also be increased for the same reason. In figures 9, 10 and 11, the unit of the conductivity data is missing in y-scale.

  1. Authors choose to call a literature reference with the name of the first author of the publication, for instance "Coletta et al.". However, Coletta is the first author of references 28, 29 and 30. Authors should find a way to remove this ambiguity. Moreover, "Coletta" is misspelled as "Colette" at two places.

4. Authors should decide of a homogeneous format for their references. For instance, in references 28, 29 and 30, authors' formats are respectively "first name + surname", "surname + initial" and "initial + surname". In reference 28, date is between brackets and after the authors' list, while date is without brackets and after the journal name in reference 30.
